# Cooperative transport mechanism of human monocarboxylate transporter 2

Bo Zhang [1,10], Qiuheng Jin [1,10], Lizhen Xu[2,10], Ningning Li [3,10], Ying Meng[4], Shenghai Chang[5,6], Xiang Zheng[7], Jiangqin Wang[5,8], Yuan Chen[7], Dante Neculai [4], Ning Gao [3], Xiaokang Zhang [9✉], Fan Yang[2✉], Jiangtao Guo [5,8✉] & Sheng Ye [1,9✉]

Proton-linked monocarboxylate transporters (MCTs) must transport monocarboxylate efficiently to facilitate monocarboxylate efflux in glycolytically active cells, and transport monocarboxylate slowly or even shut down to maintain a physiological monocarboxylate concentration in glycolytically inactive cells. To discover how MCTs solve this fundamental aspect of intracellular monocarboxylate homeostasis in the context of multicellular organisms, we analyzed pyruvate transport activity of human monocarboxylate transporter 2 (MCT2). Here we show that MCT2 transport activity exhibits steep dependence on substrate concentration. This property allows MCTs to turn on almost like a switch, which is physiologically crucial to the operation of MCTs in the cellular context. We further determined the cryo-electron microscopy structure of the human MCT2, demonstrating that the concentration sensitivity of MCT2 arises from the strong inter-subunit cooperativity of the MCT2 dimer during transport. These data establish definitively a clear example of evolutionary optimization of protein function.

[1] Life Sciences Institute, Zhejiang University, Hangzhou 310058, China. [2] Department of Biophysics and Kidney Disease Center, First Affiliated Hospital, Institute of Neuroscience, NHC and CAMS Key Laboratory of Medical Neurobiology, Zhejiang University School of Medicine, Hangzhou 310058, China. [3] State Key Laboratory of Membrane Biology, Peking-Tsinghua Center for Life Sciences, School of Life Sciences, Peking University, Beijing, China. [4] Department of Cell Biology, School of Basic Medical Sciences, Zhejiang University, Hangzhou, Zhejiang, China. [5] Department of Biophysics, Department of Pathology of Sir Run Run Shaw Hospital, Zhejiang University School of Medicine, Hangzhou 310058, China. [6] Center of Cryo Electron Microscopy, Zhejiang University School of Medicine, Hangzhou 310058, China. [7] The State Key Laboratory of Subtropical Silviculture, Zhejiang A & F University, 666 Wusu street, Lin'an 311300, China. [8] Department of Biophysics, Institute of Neuroscience, NHC and CAMS Key Laboratory of Medical Neurobiology, Zhejiang University School of Medicine, Hangzhou 310058, China. [9] Tianjin Key Laboratory of Function and Application of Biological Macromolecular Structures, School of Life Sciences, Tianjin University, 92 Weijin Road, Nankai District, Tianjin 300072, China. [10] These authors contributed equally: Bo Zhang, Qiuheng Jin, Lizhen Xu, Ningning Li. ✉email: xzhang1965@zju.edu.cn; fanyanga@zju.edu.cn; jiangtaoguo@zju.edu.cn; sye@tju.edu.cn

Pyruvate and lactate are the end products of glycolysis and major substrates for the tricarboxylic acid (TCA) cycle. They constitute critical branch points in cellular metabolism, lying at the intersection of catabolic pathways with anabolic pathways for lipid synthesis, amino acid biosynthesis, and gluconeogenesis. Indeed, dysregulated carbon metabolism becomes emerging hallmarks of diabetes, obesity, and cancer[1–3]. Under aerobic conditions, glucose was generally assumed to be fully catabolized to carbon dioxide in cells via the concerted action of glycolysis and the TCA cycle. However, in recent years, there is an increased awareness of the metabolic flexibility, in which glycolysis and the TCA cycle are uncoupled, thus allowing independent cell-specific regulation of both processes[4]. Notably, circulating lactate was recently demonstrated to be a major source of carbon for the TCA cycle both in normal and cancerous tissues[5], highlighting the physiological significance of monocarboxylate transport.

In human, the rapid exchange of both cellular lactate and pyruvate with the circulation is mainly mediated by MCTs encoded by the SLC16 family members[6,7]. The aim of this study is to address two fundamental questions surrounding this process. The first question is related to the cellular environment in which MCTs operate. When the cellular lactate concentration is high, such as that in glycolytically active cells, MCTs transport monocarboxylate efficiently to facilitate lactate efflux, maintaining an intracellular pH and monocarboxylate homeostasis[7]. While in glycolytically inactive cells with low cellular monocarboxylate concentration, MCTs transport monocarboxylates slowly or even shut down to maintain a physiological monocarboxylate concentration inside the cell, especially that of pyruvate. Maintaining cellular pyruvate above a minimum concentration is important to ensure that pyruvate is reduced to L-lactate, regenerating cytosolic $NAD^+$ from NADH and thus allowing glycolysis to continue[8]. To understand the properties of MCTs, we should first address whether MCTs can adjust their transport activities in response to monocarboxylate concentration. If the answer for the first question is yes, then the second question is how MCTs sense the difference in monocarboxylate concentration.

To address the above questions, we performed pyruvate transport assay and cryo-EM structural analysis on MCT2. The cryo-EM structure, as well as the detailed functional assay, reveal a striking switch-like behavior arising from the strong intersubunit cooperativity of the MCT2 dimer during transport, and provide insights into substrate recognition, energy coupling, and the transport mechanism of MCT2.

## Results

**MCT2 transports pyruvate cooperatively.** We first confirmed the localization of mEGFP tagged MCT2 to the plasma membrane without co-expression of embigin in HEK293 cells by confocal imaging (Supplementary Fig. 1a), as embigin was reported necessary[9,10] or unnecessary[11] for the intracellular trafficking of the MCT2 to the plasma membrane. Next, HEK293T cells co-transfected with pyronic, a genetically encoded pyruvate FRET sensor[12], and either empty vector control or MCT2, were exposed to pyruvate, resulting in an increase of the FRET-based fluorescence signal indicative of pyruvate uptake (Fig. 1a, Supplementary Fig. 1, Supplementary Movie 1). This signal is completely eliminated upon pre−incubation of the cells with AR-C155858, an inhibitor of MCTs[13] (Fig. 1a), confirming that the signal in control cells arises from endogenous MCTs-dependent pyruvate transport. Comparing the influx (rising phase) and efflux (falling phase) rates of pyruvate transport in MCT2-expressing and control cells, we observed that the rates in

MCT2-expressing cells are more than 60% higher in both directions than that in control cells. (Fig. 1b). When pyronic was replaced with a pH-sensitive fluorescent probe, BCECF-AM[14], following addition of pyruvate, we observed an ~150% increase in the rate of acidification, reciprocal to influx of pyruvate (Fig. 1c, d). These data confirm an $H^+$-coupled pyruvate transport of MCT2.

Next, HEK293T cells co-transfected with pyronic and MCT2 were bathed in 10 mM pyruvate until equilibration, raising the intracellular pyruvate concentration, and were then exposed to buffer without pyruvate to record the FRET-based fluorescence signal as a function of time (Supplementary Fig. 2a, b, Supplementary Movie 2). The FRET-based fluorescence signal of each recording point was converted to an individual pyruvate concentration based on the correlation reported previously (Supplementary Fig. 2c, d)[12], and the corresponding transient pyruvate efflux rate was calculated. Fig. 1e shows the pyruvate efflux rate measured at an extracellular pyruvate concentration of 0 mM as a function of intracellular pyruvate concentration (Supplementary Fig. 2e). The efflux rate increases slowly at low pyruvate concentrations (moderate regime), begins to tip up at approximately 40 μM, and then continues to increase in a nearly linear fashion (linear regime). The efflux rate−concentration curve reveals a transition from moderate to linear regimes. Such a transition represents a critical concentration of substrate for MCT2, below which the transporter barely operates. To perceive the transport mechanism, we define MCT2 transport activity as the pyruvate efflux rate versus the intracellular pyruvate concentration, and then plotted a pyruvate dose-response curve (Fig. 1f, Supplementary Fig. 2f). Such a sigmoidal activity-concentration curve reveals a steep dependence of transport activity on substrate concentration, and can be described by the Hill equation, with a Hill coefficient ($n$ value) of 1.6, indicating strong positive cooperativity in transport (Fig. 1f, g). These data demonstrate that MCTs may turn on almost like a switch in response to subtle difference in monocarboxylate concentration, which is crucial to the physiological operation of MCTs in the cellular context.

**Structure determination and overall structure of MCT2.** To understand the cooperative transport mechanism, we determined the cryo-EM structure of MCT2 at a resolution of 3.8 Å. The density map reveals a homodimeric architecture of MCTs and a well-resolved transmembrane domain (TMD) with clear visible α-helical features (Fig. 2a, Supplementary Figs. 3 and 4). The good quality TMD density enabled the building of a molecular model (Fig. 2b) that included near all side chains for all the transmembrane helices (TMs) together with most of the loops between TMs. However, majority of the large putatively cytoplasmic loop between TM6 and TM7 was not modeled due to poor density, which is highly sensitive to proteolytic degradation in membrane preparations[15]. Nevertheless, a short intracellular α-helix (ICH) in this loop has clear density and was modeled (Fig. 2b, Supplementary Fig. 4).

Consistent with an earlier prediction[15], MCT2 contains 12 TMs, with both the N and C termini located on the intracellular side (Fig. 2c). However, it is noteworthy that 6 of the 12 TMs, including TM1, TM2, TM5, TM7, TM8 and TM10, are discontinuous helices (Fig. 2c, Supplementary Fig. 5), which may facilitate conformational changes during substrate transport[16]. In addition, TM12 is unusually long and extends into cytoplasm. Similar to known structures of MFS transporters[17–19], the 12 TMs are organized into two six-helix bundle domains (TM1-6 and TM7-12). The two domains share a similar arrangement and are related by a pseudo-two-fold symmetry

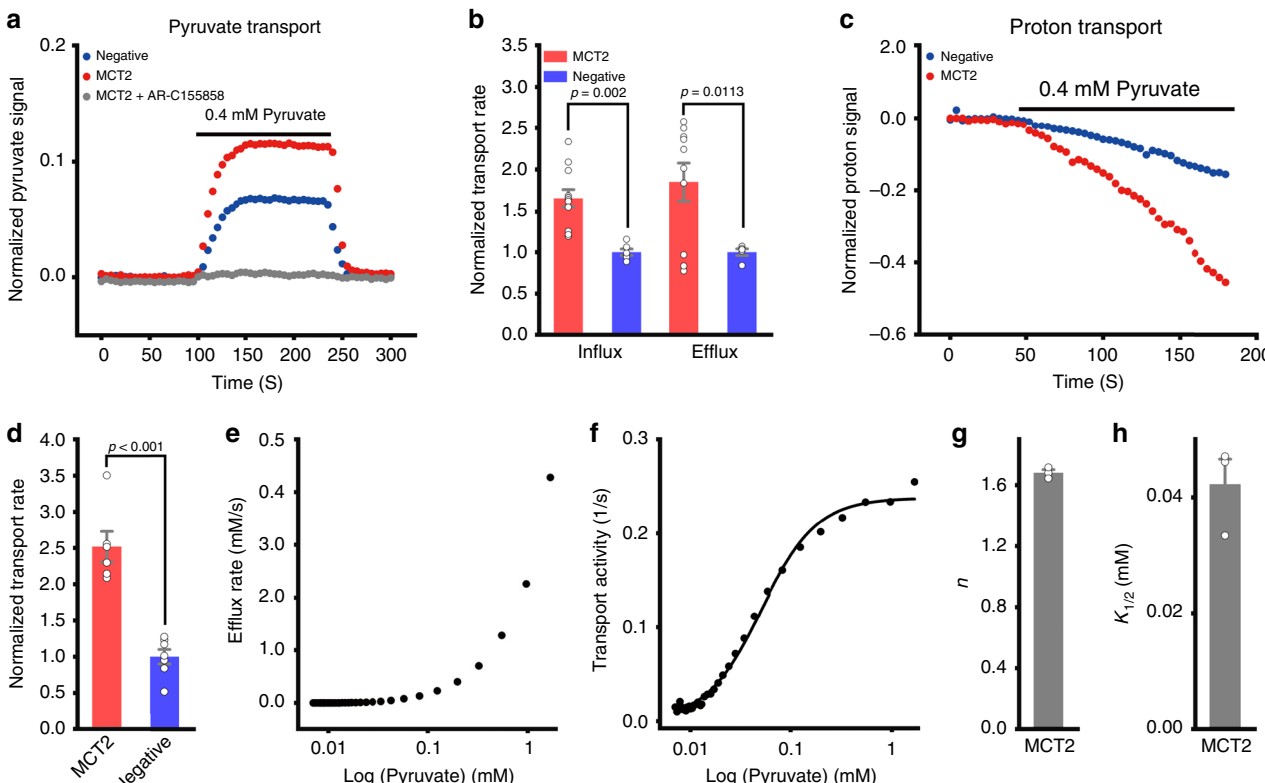

**Fig. 1 MCT2 transports pyruvate cooperatively. a–d** Assessment of pyruvate (pyronic[12]) and proton (BCECF-AM[14]) flux in HEK293T cells expressing either MCT2 (red) MCT2 upon pre-incubation with AR-C155858[13](gray), or empty vector control (blue). **a** Representative traces and b, bar plots depicting normalized rates of pyruvate influx and efflux (for pyruvate influx, WT MCT2, $n = 12$; Negative, $n = 6$; for pyruvate efflux, WT MCT2, $n = 10$; Negative, $n = 5$). **c** Corresponding representative traces and **d** bar plots depicting normalized rates of proton influx (WT MCT2, $n = 6$; Negative, $n = 7$). **e, f** Plots of **e**, pyruvate efflux rate or **f** transport activity measured at an extracellular pyruvate concentration of 0 mM as a function of intracellular pyruvate concentration for MCT2 at an extracellular pH 7.4. The smooth lines are fitted with the Hill equation, activity=activity$_{max}$/(1 + (K$_{1/2}$/[pyruvate])$^n$), where n is the Hill coefficient and K1/2 is the [pyruvate] required for activity to reach half of maximum. **g, h** Quantification of Hill coefficient ($n$ value) and K1/2 of WT MCT2. Bar plots of **g** depicting Hill coefficient (n value) ($n = 3$). Bar plots of h depicting K$_{1/2}$ ($n = 3$). For WT MCT2, $n = 1.6$ and K1/2 = 42 μM. Bar plots data were presented as mean±SEM. One-tailed $t$-tests.

axis that is perpendicular to the membrane bilayer (Supplementary Fig. 6b). Each domain comprises a pair of internal structural repeats related by an approximate 180° rotation around an axis parallel to the membrane bilayer (Supplementary Fig. 6b). The two domains contact at the extracellular side with this side tightly closed. Consequently, the MCT2 structure assumes an inward-open conformation, creating a large cavity that is continuous only with the intracellular side (Fig. 2d). This cytosolic-facing cavity, situated approximately halfway across the membrane bilayer, mainly formed by TM1, TM2, TM5, TM7, TM8, TM10 and TM11, has a narrow intracellular entrance.

**Subunit cooperativity underlies a cooperative transport**. MCT2 exists as a homodimer. Analysis of the dimer interface reveals distinctive features. First, the two subunits in the MCT2 dimer bury an extensive interface of 5100 Å[2] involving 4 TMs from each subunit, and are related by a dyad perpendicular to the membrane (Supplementary Fig. 7). On the periphery of the interface, hydrophobic amino acids from TM8 of one MCT2 subunit interdigitate with nonpolar residues from TM6 and TM1 of the adjacent subunit, whereas on the center, two TM5s from both subunits cross over (Fig. 3a). Second, two defining signature motifs across all SLC16 family are involved in inter-subunit interactions (Fig. 3b). The first one, with the sequence of [15]DGGWGW[20] in MCT2 (Supplementary Fig. 3), traverses the lead into TM1, while the

second one, with that of [138]YFYRKRPMANGLAMAG[153] in MCT2, constitutes the loop between TM4 and TM5 and the beginning of TM5[20]. The N-terminus of TM1 from one subunit extends into a pocket formed by TM5, TM8 and a loop between TM4 and TM5 of the adjacent subunit (Fig. 3b). Key residues from two motifs form direct inter-subunit interactions. For example, the guanidine moiety of Arg143, a conserved residue from the second motif, forms a cation-π interaction with the indole ring of Trp18 from the first motif of the adjacent subunit, while Asn147 forms a hydrogen bond with the main chain amide group of Trp20 from the adjacent subunit (Fig. 3c). Third, both N- and C-terminal domains are involved in dimerization (Fig. 3a). This is exceptional in MFS transporters, as they shuttle substrate across cell membranes using an alternating-access mechanism, involving domain rotation and local structural rearrangement of the two domains[21]. Taken together, the extensive dimer interface, the coupling between two defining signature motifs, and the participation of both domains in dimerization, suggest that dimerization is functionally required and both subunits work cooperatively in transporting substrate.

To investigate the fundamental question of whether the concentration sensitivity of MCT2 arises from the strong inter-subunit cooperativity during transport, we individually mutated six key residues at the dimer interface, Trp18 and Trp20 from the first signature motif, Arg143 and Asn147 from the second one, and Asn305 and Glu360, two residues forming a hydrogen bond

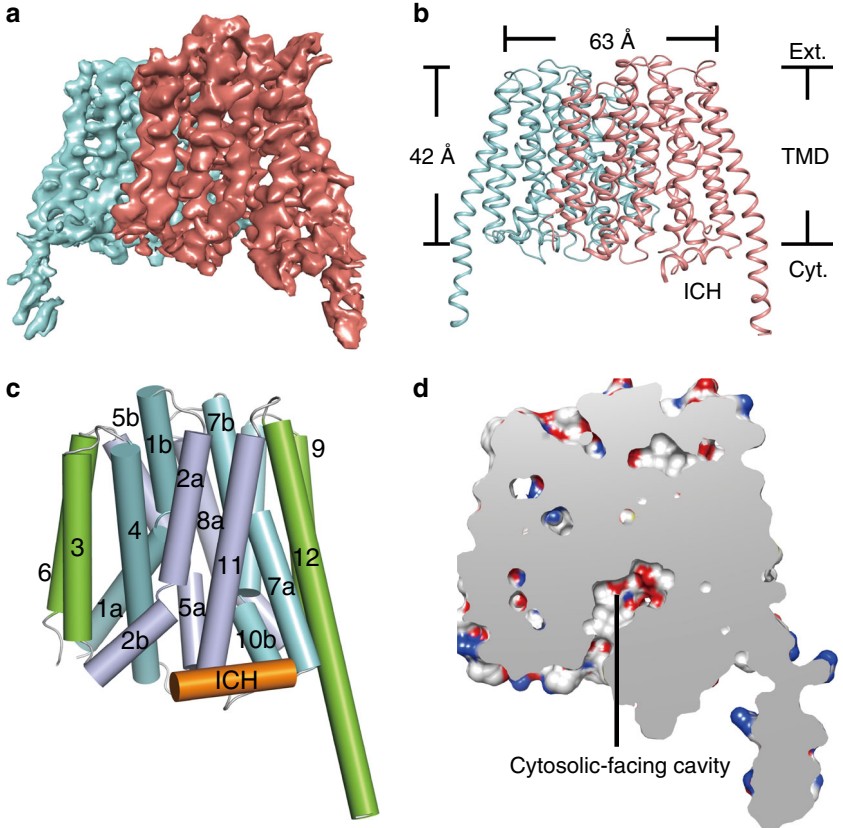

**Fig. 2 Structure of human MCT2. a** A 3D reconstruction of MCT2 with each subunit individually colored in cyan and magenta. **b** Cartoon diagrams of MCT2 dimer in the same orientation as the electron microscopy map in **a**. Ext, TMD and Cyt mean extracellular side, transmembrane domain and cytoplasmic side. **c** The structure of MCT2 subunit in an inward-open conformation viewed parallel to the membrane. The corresponding transmembrane segments in the four 3-helix repeats are colored the same. The intracellular helix is colored as orange, respectively. **d** A slab of cut-open view of the surface is shown to facilitate visualization of the inward-facing cavity.

that are very close to Try18 (Fig. 3c), to alanine and characterized their transport phenotypes. We observed substantially reduced transport activities for all six mutants (Fig. 3d), without significant differences on protein expression level and cell surface localization (Supplementary Fig. 10a, c). Given the extensive dimer interface (Supplementary Fig. 7), the reduced activities are unlikely caused by the disrupted dimer interface. Notably, three of them (W18A, W20A and R143A) were dominant-negative, with the activities even lower than that of negative control (Fig. 3d). As the mutant forms a mixed population of wt/mutant dimer with the endogenous MCT2 in addition to homodimers of wt/wt and mutant/mutant, the observed dominant negative effect indicates that mutation on one subunit does affect the activity of the other one within a dimer, supporting a cooperative transport. These data are consistent with previous observations that mutations on Arg143 resulted in complete inactivation of the MCT1[22]. We further chose R143A MCT2, a mutant with the strongest dominant-negative effect, and generated the efflux- and activity-concentration curves as those of wild type MCT2. The efflux rate-concentration curve of R143A MCT2 reveals a transition from moderate to linear regimes at approximately 180 μM, a much higher pyruvate concentration compared to that of wild type MCT2 (Fig. 3g, Supplementary Fig. 2g, h). Moreover, the activity-concentration curve of R143A MCT2 reveals a dramatically decreased Hill coefficient of 1.0, indicating loss of cooperativity, consistent with the dominant-negative effect we observed (Fig. 3d, e, f, Supplementary Fig. 2i). The results demonstrate that strong cooperativity exists between two subunits of a MCT2 dimer underlying a cooperative transport.

**Implications for substrate recognition and proton coupling.** MCTs exhibit a broad specificity for short chain mono-carboxylates including pyruvate and L-lactate. While the current resolution of 3.8 Å was insufficient to determine the complex structures of MCT2 and the small substrates, we took an alternative approach by molecular docking to probe the conformation of the pyruvate-MCT2 complex (Methods). Among the 110400 decoys generated, the top ten docking models with lowest binding energies exhibited good structural convergence. The carboxylate group of pyruvate points toward the guanidine moiety of Arg297 and the hydroxyl groups of Tyr34 and Ser355, while the methyl group interacts with Phe351 (Fig. 4b). The docking results supported the existing findings for the substrate recognition[22–24]. Arg297 and Phe351 are highly conserved across MCTs. Arg297 (Arg306 in MCT1) had been suggested to be directly involved in substrate recognition in MCT1[25]. And mutation of Phe360, the corresponding residue of Phe351 in MCT1, to cysteine shifts the substrate specificity to mevalonate, a large monocarboxylate which is not a substrate for wild-type MCT1[26,27].

To further confirm the docking results, we chose seven residues, Tyr34, Lys38, Arg297, Phe351, Ser355, Tyr70 and Phe262, that are closed to substrate in the docking models, to generate point mutants in MCT2, and characterized their transport phenotypes (Fig. 4b). While all these mutants maintain similar expression level and cell surface localization (Supplementary Fig. 10b, c), mutations on Tyr34, Lys38, Arg297, Phe351 and Ser355 resulted in substantially reduced transport activities, while those on Tyr70 and Phe262 didn't, consistent with our docking results. It is worth noting that Ser355 is only conserved in MCT1

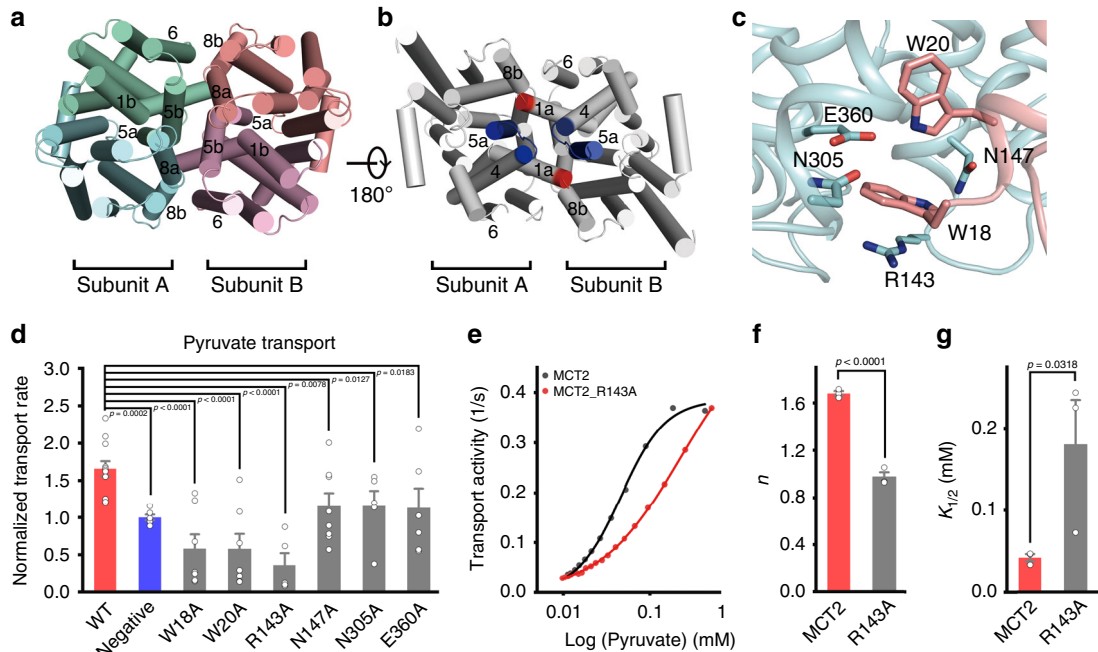

**Fig. 3 Subunit cooperativity of MCT2 supports a cooperative transport. a**, **b** Cartoon representations of MCT2 dimer viewed from **a** extracellular and **b** intracellular sides. **a** The N- and C-terminal six-helix bundles of the first subunit are colored in green and cyan, whereas those of the second subunit in pink and red, respectively. **b** The two defining signature sequences of SLC16 family are colored in red and blue, whereas the others in gray, respectively. **c** Zoom in view of the dimer interface showing the interactions between the two defining signature sequences from different subunits. The first subunit is colored in cyan, whereas the second subunit in red, respectively. **d** The pyruvate influx rates of HEK293T cells transformed with MCT2 variants. Wild type (WT) MCT2 and empty vectors were used as positive and negative controls. Bar plots depicting normalized rates of pyruvate influx (WT MCT2, $n = 12$; Negative, $n = 6$; W18A, $n = 7$; W20A, $n = 7$; R143A, $n = 5$; N147A, $n = 8$; N305A, $n = 5$; E360A, $n = 6$). Details of the experiments can be found in Supplementary Methods. **e** Transport activity measured at an extracellular pyruvate concentration of 0 mM as a function of intracellular pyruvate concentration for WT or R143A MCT2s at an extracellular pH 7.4. The smooth lines are fitted with the Hill equation, activity=activity$_{max}$/(1 + (K$_{1/2}$/ [pyruvate])$^n$), where n is the Hill coefficient and K$_{1/2}$ is the [pyruvate] required for efflux to reach half of maximum. **f**, **g** Quantification of Hill coefficient (n value) and K1/2 of WT and R143A MCT2. Bar plots of f depicting Hill coefficient (*n* value) (WT MCT2, $n = 3$; R143A, $n = 3$). Bar plots of g depicting K$_{1/2}$ (WT MCT2, $n = 3$; R143A, $n = 3$). For WT MCT2, $n = 1.6$ and $K_{1/2} = 42$ μM; for R143A MCT2, $n = 1.0$ and $K_{1/2} = 180$ μM. Bar plots data were presented as mean ± SEM. One-tailed *t*-tests.

and MCT2, two MCTs exhibiting a high affinity for pyruvate[28,29]. The equivalent residue in MCT3 and MCT4 is a glycine, and correspondingly, pyruvate affinity is reduced by approximately 100-fold in MCT4[30]. Given the similar lactate affinity between MCT4 and MCT1, Ser355 might be key for pyruvate selectivity but not for lactate selectivity.

To explain why MCTs show strong stereoselectivity for L- over D-lactate, we performed additional molecular docking studies by placing either L- or D-lactate within the ligand binding pocket. Among the top 10 models with lowest binding energy, only the L-lactate molecule was observed, strongly suggesting that L-lactate binds with a much higher affinity. Similar to pyruvate, the carboxylate group of L-acetate forms hydrogen bonds with Tyr34, Arg297 and Ser355, respectively (Fig. 4b). While the hydroxyl group of L-acetate forms hydrogen bonds with both Tyr34 and Lys38 (Fig. 4b). Moreover, the methyl group of L-lactate points toward and contact Phe351 (Fig. 4b). This binding mode explains why MCTs only transport short chain monocarboxylates and why mutation on this residue (Phe351) shifts the substrate specificity. Since any chemical group larger than methyl would introduce a steric clash with Phe351, and mutation of Phe351 to a residue with a smaller side chain, such as cysteine, would allow the accommodation of a large monocarboxylate, such as mevalonate[26,27].

To understand the H⁺-coupling mechanism, we first focused on identification of the potential residues involved in H⁺ binding. By superimposing MCT2 with lactose permease (LacY)[17], a H⁺-coupling lactose transporter, we observed that

Asp293 in MCT2 is closed to Glu325 in LacY, a residue playing a critical role in proton coupling. In addition, Asp293 is embedded in a hydrophobic milieu formed by Val156, Met289, Ala290 and Phe351 (Fig. 4c). Moreover, Asp293 is strictly conserved in all identified proton-linked monocarboxylate transporters (MCTs) encoded by the SLC16 family members (Supplementary Fig. 5), and forms a charge pair with Arg297, a residue directly involved in substrate recognition (Fig. 4c). We then created a neutral replacement mutant (D293N) for Asp293, characterized its transport phenotype (Fig. 4e) and assessed whether it utilizes an H⁺-coupled transport mechanism (Fig. 4f). The mutation D293N in MCT2 let to substantially reduced proton-dependent active symport (Fig. 4f), but not counterflow activity (Fig. 4e).

Earlier studies on MCT1 had suggested that three strictly conserved residues, Lys38 in TM1, Asp302 and Arg306 in TM8 (Lys38, Asp293 and Arg297 in MCT2), play key roles in mediating monocarboxylate transport[23–25]. Lys38 was identified as an exofacial residue when MCT1 was in an outward-open conformation[24]. While in the inward-open MCT2 structure, the side chain of Lys38 locates in a hydrophobic environment at the bottom of the inward-open cavity. It is intriguing that the MCT2 structure shows Tyr34 to be within H-bond distance of Lys38, Asp293 and Arg297, mediating a hydrogen bond network between these key residues (Fig. 4c). Notably, elimination of the hydroxyl group of Tyr34 (Y34F), or mutations of Lys38, all abolished active transport (Fig. 4d), indicating the importance of this hydrogen network.

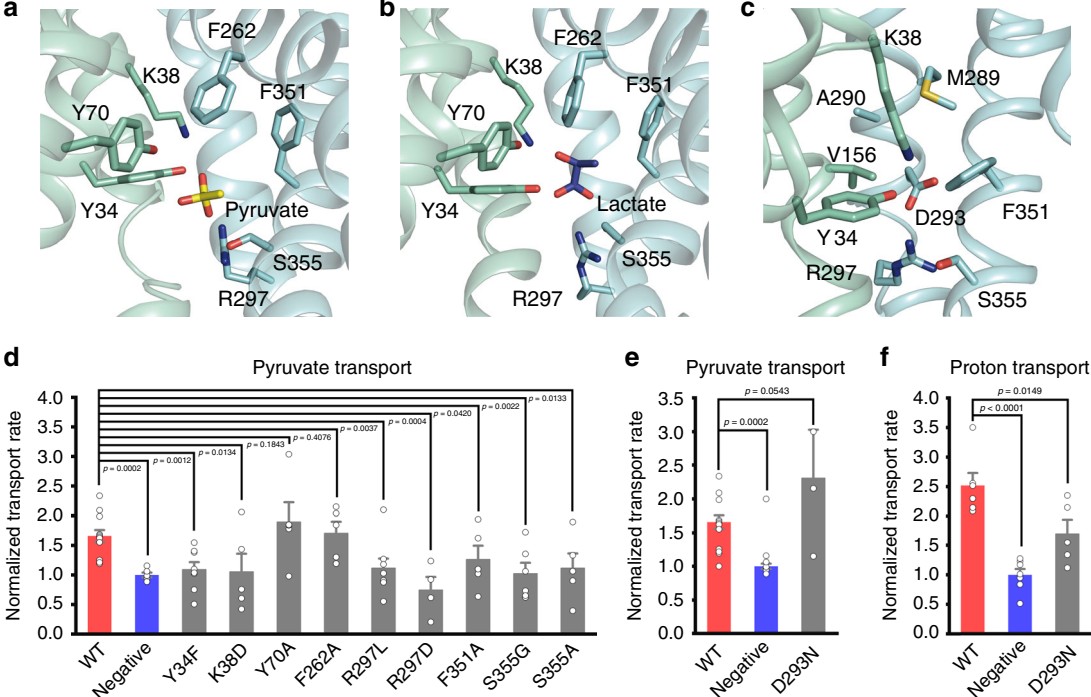

**Fig. 4 Implications for substrate recognition and proton coupling. a**, **b** Binding configurations of a, pyruvate or b, L-lactate in the MCT2 central cavity from docking studies. Details of the docking experiments can be found in Supplementary Methods. **c** Zoom in view of the MCT2 central cavity showing the residues around Asp293. **a–c** The protein is colored in cyan with corresponding residues and substrates shown in stick. **d** The pyruvate influx rates of HEK293T cells transformed with MCT2 variants. Wild type (WT) MCT2 and empty vectors were used as positive and negative controls. Bar plots depicting normalized rates of pyruvate influx (WT MCT2, $n=12$; Negative, $n=6$; Y34F, $n=8$; K38D, $n=5$; Y70A, $n=5$; F262A, $n=5$; R297L, $n=8$; R297D, $n=4$; F351A, $n=5$; S355G, $n=6$; S355A, $n=5$). e, The pyruvate or f, the proton influx rates of HEK293T cells transformed with MCT2 variants. Bar plots depicting normalized rates of pyruvate or proton influx (for pyruvate influx, WT MCT2, $n=12$; Negative, $n=6$; D293N, $n=3$; for proton influx, WT MCT2, $n=6$; Negative, $n=7$; D293N, $n=5$). Bar plots data were presented as mean ± SEM. One-tailed $t$-tests.

## Discussion

In this study, we used pyronic[12] and MCT2-transduced HEK293 cells to quantitatively study pyruvate transport, which allows us to observe the cooperative nature of MCT2. However, such a cell-based system suffers from potential interferences by endogenous MCT1 and pyruvate metabolism. We took two approaches to reduce the interferences. HEK293 cells contain endogenous MCT1 and MCT2[6]. By taking advantages of the difference between Michaelis constant (Km) values of MCT1 and MCT2 for pyruvate, majority of the data were recorded with intracellular pyruvate concentration lower than the Km value of MCT1 (1.0 mM)[6], while significantly higher than that of MCT2 (0.08 mM)[6], to ensure that under our experiments, MCT2 is fully operational, while MCT1 is not. Indeed, our data reveal that several MCT2 mutants (R297D, W18A, W20A, R143A) show pyruvate transport activities significantly lower than that of negative control, indicating the negligible contribution of MCT1 to the negative control signal. Pyruvate transport is affected by pyruvate metabolism, including mitochondria, lactate dehydrogenase (LDH) and amino-transferases. In addition, pyronic only monitors the intracellular pyruvate concentration, while MCT2 transports multiple mono-carboxylates[28]. By recording pyruvate efflux rate, all transportable monocarboxylates of MCTs are transported outward. Their efflux rates and their intracellular concentrations, are likely proportional to each other. In this perspective, the pyruvate efflux data reflect an overall monocarboxylate transport by MCTs. Indeed, our data allow us to calculate a Hill coefficient (n value) about 1.6 for wild type MCT2, and that about 1.0 for R143A MCT2, indicating negligible interferences of pyruvate metabolism. However, to further quantitatively study monocarboxylate transport in future, an

in vitro system without the interferences by other MCTs and metabolism need to be developed.

A bacterial homolog of MCT from Syntrophobacter fumar-oxidans (SfMCT) that shares sequence identity of 22% and similarity of 50% with human MCT2 was recently identified (Supplementary Fig. 8), and the crystal structures of SfMCT in its outward-open, monomeric state were determined[31]. Structural alignment of the two closely related MCTs in two distinct con-formations (Supplementary Fig. 9a, b) reveals the molecular basis for the alternating access cycle. First, transition from inward-open to outward-open states is achieved by an approximately 30° concentric rotation of the two domains that closes the intracel-lular gate by forming interactions between TM2, 4, 5 and TM8, 10, 11 (Supplementary Fig. 9f), and opens the extracellular gate by outward motions of TM1, 2, 5 and TM7, 8, 11 (Supplementary Fig. 9e). Second, both N- and C-terminal six-helix bundle domains share structural conservation between MCT2 and SfMCT, with r.m.s. deviations of 2.1 Å for 138 Cα atoms of the N-terminal domain, and 2.4 Å for 175 Cαs of the C-terminal one. Third, TM1, 5 in the N-terminal domain, and TM8, 10, 11 in the C-terminal domain undergo prominent local structural rearrange-ment during the state transition (Supplementary Fig. 9c, d).

We next generated an outward-open model of MCT dimer by superimposing SfMCT monomer on each subunit of MCT2 dimer and observed that majority of the TMs with local structural rearrangement during the state transition, including TM1, 5 and 8, locate at the dimer interface (Supplementary Fig. 9g). In the context of MCT dimer, this indicates that they are involved in the crosstalk between two subunits of a MCT2 dimer, thus resulting in two effects. First, substrate binding in one subunit of a MCT2

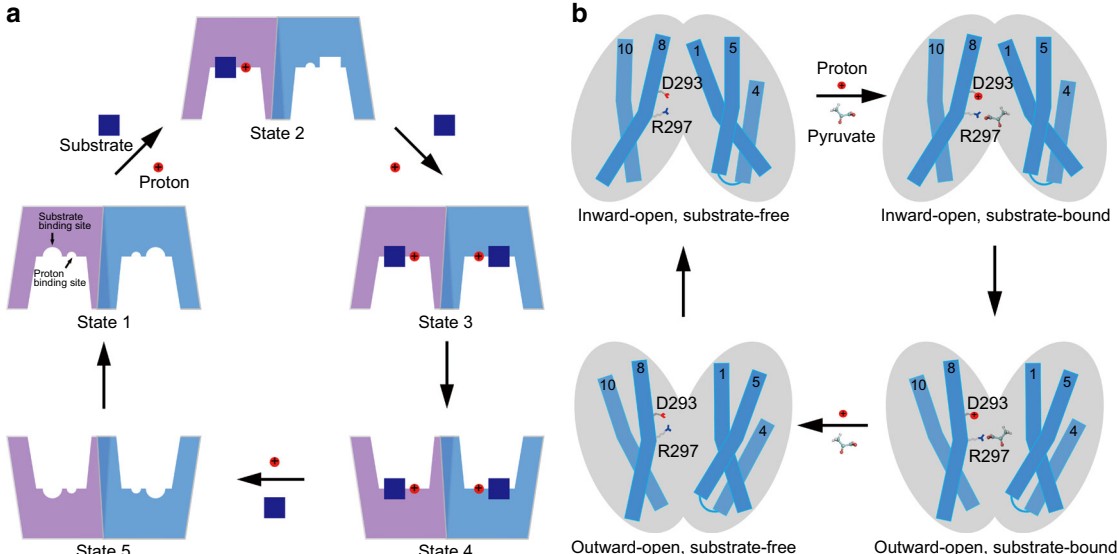

**Fig. 5 Proposed working models depicting the transport mechanism of MCT2. a** Proposed cooperative transport mechanism of MCT2. Shown here are the allosteric MCT2 dimer in five different states: inward-open, low affinity, substrate-free state (state 1), inward-open, high affinity state with substrate-bound only in a subunit (state 2), inward-open, high affinity state with substrate-bound (state 3), outward-open, high affinity state with substrate-bound (state 4), outward-open, low affinity, substrate-free state (state 5). The details of the models are discussed in the main text. **b** Proposed monocarboxylate/H$^+$ symport mechanism of MCT2. Shown here are the predicted conformations of MCT2 subunit: inward-open, substrate-free and -bound, and outward-open, substrate-bound and -free, required for a complete transport cycle according to the alternating access model. The inward-open substrate-free conformation of MCT2 is reported in this study.

dimer may drive the conformational change of the substrate binding site, altering the ligand-binding affinity in adjacent subunit. Second, both subunits are coupled, changing their conformational states in a concerted manner.

The data presented here have allowed us to propose a simplified model of cooperative transport in MCT2 (Fig. 5a). The MCT2 structure was captured in an inward-facing conformation (State 1, Fig. 5a). The cytosolic-facing cavity observed in the structure contains a substrate- and a proton-binding sites. Asp293 is a potential proton-binding residue, while four residues from three TMs, including Arg297 from TM8, Tyr34 from TM1, Phe351 and Ser355 from TM10, together form a substrate-binding site (inward-open, substrate-free state, Fig. 5b). Based on our functional studies, the substrate-binding site in this state likely exhibits a low affinity for monocarboxylates. Asp293 and Arg297 play an essential role to couple proton translocation and substrate recognition. Upon protonation of Asp293, the proton neutralizes the negative charge of Asp293, loosens the salt bridge between Asp293 and Arg297, and lowers the energetic barrier for monocarboxylate binding and/or transport (inward-open, substrate-bound state, Fig. 5b). The monocarboxylate binding in one subunit of a MCT2 dimer involves Tyr34 from TM1 and Asp293 and Asp297 from TM8, which would consequently induce movement of TM1 and TM8. Given the evolutionary optimized coupling between TM1 from one subunit and the TM4-5 loop of adjacent subunit (Fig. 3a, b), substrate-induced motion originating in one subunit is then transmitted to the other via an extensive network of interactions at the dimer interface. These conformational changes may alter the ligand-binding affinity in adjacent subunit (State 2, Fig. 5a), resulting in a quick ligand-binding in adjacent subunit (State 3, Fig. 5a). Upon substrate binding and protonation of both subunits, an inward-to-outward transition of MCT2 dimer occurs (State 4, Fig. 5a, outward-open, substrate-bound state, Fig. 5b). The transport is complete with the release of the substrate (State 5, Fig. 5a, outward-open, substrate-free state, Fig. 5b) and an outward-to-inward transition back to state 1 for next cycle. The proposed

models fit the functional observations and are supported by structural analyses. Moreover, the structural and functional studies reported here provide insights into the function of MCTs and serve as a framework for future investigation.

## Methods

**Constructs, protein expression and purification.** The full-length human SLC16A7 gene (NCBI accession NM_001270622) containing a C-terminal Strep tag was cloned into a pEZT-BM vector[32] for heterologous expression and western blot analysis. The gene was also cloned into a pEG-BacMam vector with a mEGFP tag for cell surface expression analysis with confocal microscopy (Supplementary Table 2).

For protein expression, Strep tagged MCT2 was heterologously expressed in HEK293F cells (Thermo Fisher Scientific; R79007) using the BacMam system (Thermo Fisher Scientific). The baculovirus was generated in Sf9 cells (ATCC; CRL-1711) following standard protocol and was used to infect HEK293F cells at a ratio of 1: 10 (virus: HEK293F, v:v), supplemented with 10 mM sodium butyrate to boost protein expression.

Cells were cultured in suspension at 37 °C for 48 hours and harvested by centrifugation at 3,000 x g. The cell pellet was re-suspended in buffer A (50 mM HEPES-NaOH pH 7.5 and 150 mM NaCl) supplemented with a protease inhibitor cocktail (2 μg ml$^{-1}$ DNase I, 2 μg ml$^{-1}$ pepstatin, 2 μg ml$^{-1}$ leupeptin and 2 μg ml$^{-1}$ aprotinin, and 1 mM PMSF) and homogenized by sonication on ice. MCT2 was extracted with 2% (w:v) n-Dodecyl-β-D-Maltopyranoside (DDM, Anatrace) supplemented with 0.2% (w:v) cholesteryl hemisuccinate (CHS, Sigma Aldrich) by gentle agitation for 3 hours on ice. After extraction, the supernatant was collected following a 40-minute centrifugation at 48,000 x g and incubated with Strep-Tactin Sepharose resin (IBA) with gentle agitation. After 1 hour, the resin was collected on a disposable gravity column (Bio-Rad), washed in buffer B (buffer A + 0.1 % DDM + 0.02% CHS) for 10 column volumes and was eluted with 10 mM desthiobiotin. The protein sample was further purified by size exclusion chromatography on a Superose 6 10/300 GL column (GE Heathcare) pre-equilibrated with buffer B. The protein peak fraction was collected and concentrated to 4.0 mg/ml for cryo-electron microscopy analysis (Supplementary Fig. 3).

**EM data acquisition.** The cryo-EM grids were prepared by applying 3 μl of MCT2 to a glow-discharged Quantifoil R1.2/1.3 200-mesh gold holey carbon grid (Quantifoil, Micro Tools GmbH, Germany) and blotted for 4.0 seconds under 100% humidity at 4 °C before being plunged into liquid ethane using a Mark IV Vitrobot (FEI). Micrographs were acquired on a Titan Krios microscope (FEI) operated at 300 kV with a K2 Summit direct electron detector (Gatan), using a slit width of 20 eV on a GIF-Quantum energy filter. SerialEM software[33] was used for automated data collection, at a magnification of 165,000×, resulting in a calibrated pixel size of 0.8285 Å in super-resolution mode. The defocus range was set from −1.5 μm to −2.5 μm. Each

micrograph was dose-fractionated to 32 frames under a dose rate of 7 e⁻/pixel/s, with a total exposure time of 8 s, resulting in a total dose of about 81 e⁻/Å². 

Image processing. The motion correction was performed using the MotionCorr2 program[34], and the CTF parameters of the micrographs were estimated using the GCTF program[35]. All other steps of image processing were performed using RELION 3.0[36] and cryoSPARC 2[37]. About 1,000 particles were manually picked from a few micrographs for 2D classification. Class averages representing projections of MCT2 in different orientations were selected and used as templates for automated particle picking from the full data set of 1,653 micrographs. The particles were extracted with a binning factor of 3 and were subjected to a 2D classification. A total of 302,904 particles were selected for two rounds of 3D classifications using the initial model generated by RELION as the reference. Two of the 3D classes showed good secondary structural features and their particles were selected, combined and re-extracted into the original pixel size of 0.8285 Å. After 3D refinement with C2 symmetry, particle polishing and CTF refinement with Relion, and local-refinement refinement with cryoSPARC 2, the resulting 3D reconstructions from 100,909 particles yielded an EM map with a resolution of 3.8 Å (Supplementary Figure 3). The resolution was estimated by applying a soft mask around the protein density and the gold-standard Fourier shell correlation (FSC) = 0.143 criterion. ResMap[38] was used to calculate the local resolution map. (Supplementary Fig. 3).

Model building, refinement, and validation. De novo atomic model building based on 3.8 Å resolution density map of MCT2 was performed in Coot[39]. Amino acid assignment was achieved based on the clearly defined density for bulky residues (Phe, Trp, Tyr, and Arg). Models were refined against summed maps using phenix.real_space_refine[40], with secondary structure restraints and non-crystallography symmetry applied. The initial EM density allowed us to construct a MCT2 model containing residues 18-197 and 233-446. The statistics for the models' geometries was generated using MolProbity[41] (Supplementary Table 1). All the figures were prepared in PyMol or Chimera[42].

**Molecular docking**. RosettaLigand[43–45] application from Rosetta program suite version 3.4 was used to dock pyruvate to MCT2. Cryo-EM structure model of the human MCT2 were first relaxed in membrane environment using the Rosetta-Membrane application[46–48] and models with lowest energy scores were chosen for docking of pyruvate. Docking was comprised of three stages, which progressed from low-resolution conformational sampling and scoring to full atom optimization using all-atom energy function. In the first, low-resolution stage, pyruvate molecule was initially placed roughly in the central cavity. Pyruvate was allowed to move within a 30 Å diameter sphere, where it was randomly placed at the beginning of each docking process. Pyruvate conformers were generated using the FROG2 server[49]. The second, high-resolution stage employed the Monte Carlo minimization protocol in which the ligand position and orientation were randomly perturbed by a small deviation (0.1 Å and 3°). MCT2-contacting residue side chains were repacked using the rotamer library within the Rosetta suite. The ligand position, orientation, and torsions and protein side-chain torsions were simultaneously optimized using quasi-Newton minimization and the result was accepted or rejected based on the Metropolis criterion. The side-chain rotamers were searched simultaneously during full repack cycles and one at a time in the rotamer trials cycles. Rotamer trials chose the single best rotamer at a random position in the context of the current state of the rest of the system, with the positions visited once each in random order. The ligand was treated as a single residue and its input conformers served as rotamers during this stage. The third and final stage was a more stringent gradient-based minimization of the ligand position, orientation, and torsions and the transporter torsion angles for both side chains and backbone.

A total of 110,000 models were generated for pyruvate docking. To determine the best docking model, these models were first screened with total energy score (Rosetta energy term name: score). Top 1000 models with lowest total energy score were selected. They were further scored with the binding energy between pyruvate and MCT2. Binding energy was calculated as the difference in total energy between the pyruvate bounded state and the corresponding apo state. Top 10 models with lowest binding energy (interface_delta_X) were identified as the candidates.

Similar approaches were taken to dock L- and D-lactate to MCT2.

**Pyruvate transport activity analysis**. HEK293T cells (ATCC; CRL-3216) were transferred to a 35 mm tissue culture dish (Corning) for 48 h prior to the experiment. These cells were transiently transfected by Lipofectamine 3000 (Life Technologies) 24 h after passing on with 1 µg plasmid DNA of pyronic and pEZT-BM, MCT2-pEZT-BM or mutant MCT2-pEZT-BM following the manufacturer's protocol, respectively. Cells were transferred to glass coverslips at least 4 h prior to the imaging experiment. Twenty minutes prior to imaging, cells on glass coverslips were equilibrated in imaging solution (130 mM NaCl, 0.2 mM EDTA and 3 mM HEPES-NaOH pH 7.4). Ten minutes before imaging, the cells on the glass coverslip were transferred in a new imaging solution to equilibrate. Pyruvate uptake was initiated by switching the buffer in the perfusion tube in a gravity-driven system (RSC-200, Bio-Logic) to the imaging solution containing 0.4 mM pyruvate at the indicated time point. The pyruvate efflux was initiated by switching the buffer without pyruvate in the perfusion tube. For AR-C155858 inhibitory effect analysis, we used the solutions containing AR-C155858 (MCE).

To measure the pyruvate efflux rate, we equilibrated cells expressing pyronic and MCT2-pEZT-BM or mutant MCT2-pEZT-BM with imaging solution

containing 10 mM pyruvate for 30 min. After perfusion with imaging solution containing 10 mM pyruvate for 80 s, pyruvate efflux was initiated by switching the imaging solution without pyruvate in the perfusion tube in a gravity-driven system to the imaging solution at the indicated time point.

To measure the changes in cytoplasmic pH by the H⁺ transport, we loaded cells with 1 mM BCECF-AM (Thermo Fisher Scientific) and initiated proton transport by perfusing the imaging solution with 0.4 mM pyruvate.

For data analysis, baseline drift due to fluorescent bleaching was calculated with the data of the first 50 s, or the last 80 s (for pyruvate efflux) of each trace. Baseline drift correction with linear regression of initial trace or final trace was performed in the Igor Pro software version 5.05 (WaveMetrics) or OriginPro 2019 (OriginLab).

Our imaging system is consisted of a Nikon ECLIPSE Ti2 microscope, a wLS LED illumination system (QImaging), IsoPlane-160 spectrometer (Princeton Instruments) and an optiMOS camera (QImaging). The FRET sensor pyronic was excited using a 420/20 nm band-pass filter; with a 455 nm dichroic mirror, all fluorescence emission above 460 nm was collected. The emission spectrum was imaged with the IsoPlane-160 spectrometer and optiMOS camera. The fluorescence intensity values at emission peaks for mTFP (at 492 nm) and Venus (at 528 nm) were measured with the ImageJ software version 1.51. After subtracting the background intensity at 492 nm and 528 nm, the mTFP/ Venus emission intensity ratio was then calculated to report the transport of pyruvate.

BCECF-AM was excited first using a 420/20 nm band-pass filter and then using a 500/20 nm band-pass filter. A pair of emission spectra images of BCECF-AM excited by these two setting was acquired every four seconds with the Ocular software version 2.0 (Photometrics). The ratio of fluorescence intensity values measured at 535 nm with two excitation settings was calculated to report the influx of protons.

**Western blot analysis**. HEK293T cells were transferred to a 6-well plate (Corning) for 24 h prior to the experiment. These cells were transiently transfected by Lipofectamine 3000 (Life Technologies) 24 h after passing on with 1 µg plasmid DNA of pEZT-BM, MCT2-pEZT-BM or mutant MCT2-pEZT-BM following the manufacturer's protocol, respectively. Total protein was extracted using lysis buffer (50 mM Tris-HCl pH 7.5, 150 mM NaCl, 1 mM MgCl and 1% NP-40) and cell extracts were denatured by boiling for 10 min in SDS loading buffer, resolved by SDS–polyacrylamide gel electrophoresis (SDS-PAGE), transferred to poly-vinylidene fluoride membranes and probed with rabbit anti-strep II (1:4000 dilution; Abacm; ab76949) or mouse anti-β-actin (1:5000 dilution; Huabio; M1210-2) antibodies. The secondary antibodies used were goat anti-rabbit immunoglobulin G (IgG) (1:2000 dilution; Sangon Biotech; D110058) or goat anti- mouse IgG (1:2000 dilution; Sangon Biotech; D110087) conjugated to horseradish peroxidase.

**Confocal imaging analysis**. HEK293 cells (ATCC; CRL-1573) were transferred to a 12-well plate (Corning) containing glass coverslips for 24 h prior to the experiment. These cells were transiently transfected with 1 µg plasmid DNA of mEGFP-pEG-BacMam, MCT2-mEGFP-pEG-BacMam or mutant MCT2-mEGFP-pEG-BacMam using Lipofectamine 3000 transfection reagent (Life Technologies) following the manufacturer's protocol, respectively. 24 h post transfection, cells were washed twice with cold PBS and fixed with 4% paraformaldehyde in PBS for 15 min at room temperature. Afterwards, cells were washed three times with cold PBS and mounted on glass slides and imaged by confocal microscopy (Zeiss 800).

**Statistical analysis**. Unless otherwise specified, experiments were repeated three times, with at least 3 cells. Error bars represent SEM. Regression and statistical analyses were carried out with the computer program OriginPro 2019 (OriginLab) and Igor Pro software version 5.05 (WaveMetrics). Differences in mean values of paired samples were evaluated with the Students t-test.

**Reporting summary**. Further information on research design is available in the Nature Research Reporting Summary linked to this article.

## Data availability
Data supporting the findings of this manuscript are available from the corresponding authors upon reasonable request. A reporting summary for this Article is available as a Supplementary Information file. Structure coordinates and cryo-EM density maps have been deposited in the protein data bank under accession number PDB 7BP3 and EMD-30143. The source data underlying Fig. 1b, d, g, h; 3d, f, g; 4d-f, Supplementary Fig. 10a-b, are provided as a Source Data file.

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

## Acknowledgements

We acknowledge the use of cryoEM instruments and computing resources in Center of Cryo-Electron Microscopy, Zhejiang University, the Cryo-EM Platform of Peking University and Center for Integrative Imaging of Hefei National Laboratory for Physical Sciences at the Microscale of University of Science and Technology of China. This work was supported in part by Ministry of Science and Technology (2016YFA0500404 to S.Y., 2018YFA0508100 to J.G., 2016YFA0501102 to X.Z., and 2016YFA0500700 to N.G), the National Natural Science Foundation of China (31525001 and 31971127 to S.Y, 31870724 to J.G., 31741067 and 31800990 to F.Y., 31600606 to X.Z., 31725007 and 31630087 to N.G., and 31700655 to N.L.), Zhejiang Provincial Natural Science Foundation (LR19C050002 to J.G, LR20C050002 to F.Y.), and the Fundamental Research Funds for the Central Universities (to S.Y., J.G., F.Y., X.Z. and N.G.). This work was also supported by the Core Facilities in Zhejiang University School of Medicine, including the Bioinformatics Computation Platform and Dr. Cheng Ma at the Protein Facility.

## Author contributions

S.Y., J.G., F.Y., X.Z. and N.G. conceived and designed this project. B.Z., Q.J. and X.Z. prepared the samples; B.Z., Q.J., N.L., S.C., X.Z., J.W., N.G., X.Z., J.G. and S.Y. performed data acquisition, image processing and structure determination; L.X. and F.Y. performed the docking studies; B.Z., Q.J., X.Z., L.X. and F.Y. performed the pyruvate transport assays. B.Z., Q.J., Y.M., and D.N. performed western blot and confocal imaging. All authors participated in the data analysis and manuscript preparation.

## Competing interests

The authors declare no competing interests.
