## [Peer Review File · Nature Communications]

Reviewers' comments:

Reviewer #1 (Remarks to the Author):

According to current knowledge MCT2 is a high affinity transporter for pyruvate and lactate. It is abundant in neurons and other oxidative cells, where it mediates the import of lactate. The handling of pyruvate by MCT2 has been much less studied.

The main findings of this article are that MCT2 is a homodimer and that the transport of pyruvate by MCT2 is cooperative. The first conclusion is based on the 3D structure of MCT2 solved at 3.8 Angstrom resolution by means of cryo-electron microscopy, and the second conclusion is based on transport studies using a fluorescent probe. I am no expert on cryo-electron microscopy, so my critique will be restricted to functional aspects. As far as I am concerned the data are clearly presented, the paper is well written and the findings are novel and relevant. The cooperative nature of MCT2 is unexpected and shall prompt a revision of current models of intercellular fueling in brain tissue. Authors are invited to consider the following suggestions, which include the realization of new experiments:

Major

1. MCT2 versus MCT1. Transport rates in MCT2-transduced cells were only twice that of "negative" control cells, which are rich in MCT1 (San Martin et al., PLoS ONE 8(2): e57712). How contaminated is the transport data by MCT1? One possibility is that approx.. 50% of the activity is MCT2 and the rest is MCT1. In that case, the functional data would be hard to interpret. Alternatively, MCT2 overexpression swamped the cellular machinery (expression, trafficking, etc) and the contribution of MCT1 became negligible, a favorable scenario supported by lower than control rates in several MCT2 mutants (R297D, W18A, W20A, R143A). Ensuring that there is no significant MCT1 contamination is important for the correct interpretation of the data. One possible strategy for that would be to use pCMBS, a non-permeant reagent that inhibits MCT1 but not MCT2. If MCT2 is the main MCT in the membrane, pCMBS will not affect pyruvate uptake. A control assay in "negative" HEK cells should show complete inhibition of pyruvate uptake by pCMBS (San Martin et al., PLoS ONE 8(2): e57712).
2. MCT2 cooperativity. Estimation of efflux parameters from the time course of pyruvate depletion rests on some assumptions that need verification. It must be shown that extracellular pyruvate is fully removed from the vicinity of cells, otherwise the second phase of slow depletion could be the trivial result of incomplete pyruvate washout. Please provide perfusion rates and volume of the chamber, and demonstrate fast and complete removal, for example by imaging an extracellular dye under the same perfusion regime. Also, it is not clear how reproducible are some measurements. How many times was the experiment in Figs. 1e and 1f carried out?
3. MCT2 cooperativity. Could the slow phase of sensor decay be due to faster bleaching of one of the chromophores? Please provide data demonstrating that bleaching is not significant under your illumination conditions.
4. MCT2 saturation. Reportedly, the zero-trans K_m of pyruvate uptake via MCT2 is in the order of 50 μ M (various works by Halestrap and colleagues). However, the current study does not show saturation of zero-trans pyruvate efflux in the range 0-1.6 mM (Suppl. Fig. 2C), which points to a K_m higher than reported for native MCT2. Why is this? Is MCT2 asymmetric? Or is this explained by the fact that the transporter was expressed without its chaperone embigin? Do HEK cells express embigin? Please discuss these issues in relation to relevant MCT2 studies in oocytes by Broer and Deitmer (Biochem J. 1999 Aug 1;341 (Pt 3):529-35.; J Biol Chem. 2011 Aug 5;286(31):27781-91)
5. Fluorescent sensor calibration. Please provide a detailed protocol whereby fluorescence data was converted into concentration and illustrate it with an example. This may be included in a supplementary figure.
6. Page 12, line 6. According to a recent re-evaluation of MCT4 kinetics using the same pyruvate biosensor, its K_m for pyruvate is 4 mM, not > 150 mM, as previously thought. Thus, the affinity of MCT4 for pyruvate is "only" 100 times lower than that of MCT2, not 2000 times. (<https://www.biorxiv.org/content/10.1101/586966v1>). In the same study the K_m of MCT4 for lactate was measured at 1 mM, which is in the same range of the K_m of MCT1 for lactate. Thus,

Ser355 may be key for pyruvate selectivity but not for lactate selectivity. The same article illustrated the perils of estimating MCT parameters using pH.

7. Potential interference by pyruvate metabolism. In addition to efflux, the depletion of pyruvate upon extracellular removal may in principle be affected by other pyruvate sinks, including mitochondria, LDH and aminotransferases. However, such interference would have caused slower accumulation relative to depletion, which was not observed. Thus, the symmetry between speeds of uptake and depletion suggests that transport dominates. This should be discussed.

Minor

8. Page 5 line 5, it should read "regenerating cytosolic NAD⁺ from NADH..."

9. Page 6 line 4, the difference seems larger, e.g. about 100%?

10. Page 6 line 10-12. Highly glycolytic cells accumulate lactate, not pyruvate. Please rephrase.

Reviewer #2 (Remarks to the Author):

In the manuscript: "Cooperative transport mechanism of the human monocarboxylate transporter 2" the authors describe the cryo-EM structure of the human monocarboxylate transporter (MCT2) in inward open conformation and analyzed its pyruvate transport activity.

MCT2 is a solute carrier (member of the SLC16 family) with a typical homodimeric assembly. The essential outcome of the transport assay is the finding that MCT2 can rapidly react to changing monocarboxylate reactions necessitating strong cooperative effects between the individual monomers. The cryo-EM structure is resolved at 3.8Å resolution with solid quality, which allowed for model building. Finally, a computational approach was used to identify residues that are potentially involved in substrate binding. Point mutations of those residues led to a decreased transport activity.

I do not feel confident to ultimately judge the quality of the biochemical and transport assays; but the EM-density looks as expected at this resolution. Unfortunately, 3.8Å is nowadays on the lowish side for cryo-EM of alpha-helical membrane proteins and I wonder whether this could not be improved a bit further to solidify the model and provide a final density. Consequently, higher resolution would also alleviate any potential concerns about substrate binding and the involved residues.

As the EM-micrographs look good and the processing seems to have worked comparably straight forward I am wondering why the authors did not further push the resolution to make the manuscript much stronger. Also, the number of particles obtained (~100k) for the good class should suffice to provide an improved resolution already.

Specific comments:

Page 8 line 1: dimer bury an extensive interface of 5,147Å²

Could the authors please doublecheck, whether this is really true? From SF7 the interface appears to be confined to a much smaller area. In any case the number is way to "accurate" at 3.8 Å resolution. I would suggest to round it up.

The manuscript requires some polishing in the wording to improve readability, before publication can be considered.

Reviewer #3 (Remarks to the Author):

Zhang et al. show a cryo-EM structure of a human monocarboxylate transporter of the SLC16 family, MCT2. After the recent publication of the crystal structure of a bacterial MCT homolog, this

first human structure is highly appreciated - even though a somewhat limited resolution of 3.8 Å was achieved and parts of the structure are missing due to flexibility. Still, the structure exhibits the inside-open conformation, i.e. the complement to the outside-open bacterial structure. The authors combine their structural data with biochemical transport assays of wildtype and mutant MCT2 to foster their structure-derived hypotheses on cooperativity of the two protomers in the dimeric MCT2 complex, and on a protonation site that they make responsible for secondary active, proton-coupled transport of pyruvate. However, unfortunately, the functional data are not convincing.

Specifically:

Major points:

1. The MCT2 constructs were expressed in a mammalian cell system with a large background pyruvate transport activity. Accordingly, expression of MCT2 increased the transport rates by a small factor of around 1.5 over background. This level might suffice to conclude on general transport functionality but appears too low to attribute true differences to certain mutants. The study fully lacks quantification of the transporter of interest at the plasma membrane. Since the number of transporters directly determines the transport rate this is essential. One cannot know whether the presented rates are directly linked to structural properties of the MCT2 variants or to expression levels.
2. The ms does not mention the accessory single transmembrane proteins basigin or embigin that are required for the intracellular trafficking of the MCTs to the plasma membrane. Altered trafficking due to affected interaction of the accessory protein with the expressed MCT2 variant would also act on the measured transport rates, i.e. a second reason for determining the plasma membrane portion of MCT2.
3. The assay readout is rather indirect. By using a FRET-based pyruvate protein sensor, the authors obtained data on the pyruvate concentration in the cytosol at the respective observation time points. This concentration, however, is not only altered by transport via the heterologously expressed MCT2 protein but, at the same time, by any cellular process that affects pyruvate concentration by metabolism (lactate dehydrogenase) and alternative transport pathways (endogenous MCT isoforms, and, probably highly relevant, by mitochondrial pyruvate transporters). Even the "pyronic" sensor itself may falsify the observed transport rates at pyruvate concentrations below 100 µM due to its own affinity to pyruvate (K_D of 107 µM!). Therefore, the conclusion on cooperativity of the MCT2 dimer is not justified; effectively, if cooperativity is given, it is a result of many involved components that cannot be resolved with this assay.
4. One possibility to resolve this issue could be to determine cooperativity in the inward direction. In their assay, the authors maximally loaded the cells with pyruvate and then monitored the changes in the pyruvate release rates with decreasing concentrations over time (with the complications of alternative routes laid out above). It should be possible to determine the uptake rates, in turn, at varying external pyruvate concentrations; of course this direction of transport would be opposite to the primary physiological pyruvate transport via MCT2. What about lactate transport, i.e. the actual MCT2 substrate?
5. From their structure analysis and modelling, the authors identified aspartate-293 as a potential protonation site close to the (equally modelled) substrate binding site of MCT2. This residue is made responsible for proton binding and the co-transport property. If this was true and the proton transport mechanism would be based on Asp293, mutation to a neutral asparagine should completely abolish pyruvate transport. Yet, pyruvate transport remained unaltered in the D293N mutant (Fig. 4e) excluding Asp293 from having that impact. Still, the authors conclude from slight, hardly significant changes in the degree of a concurrently observed pH shift (Fig. 4f) that Asp293 is a key residue, which is not conclusive.

Minor points:

6. How do the authors explain the differences in the pyruvate uptake levels in the equilibrium state in Fig. 1 (not the different rates)?

7. Cryo-EM is a method that allows for depicting different conformational states via class summation and selection of different classes. Why was only the inside-open conformation picked up in the analysis?

In conclusion, again, the novel structure in its conformation is particularly helpful. Yet, the deduced structure-function relationships regarding cooperativity and protonation appear not conclusive in the current ms.

Response to reviewers' specific comments:

Reviewer #1:

According to current knowledge MCT2 is a high affinity transporter for pyruvate and lactate. It is abundant in neurons and other oxidative cells, where it mediates the import of lactate. The handling of pyruvate by MCT2 has been much less studied.

The main findings of this article are that MCT2 is a homodimer and that the transport of pyruvate by MCT2 is cooperative. The first conclusion is based on the 3D structure of MCT2 solved at 3.8 Angstrom resolution by means of cryo-electron microscopy, and the second conclusion is based on transport studies using a fluorescent probe. I am no expert on cryo-electron microscopy, so my critique will be restricted to functional aspects. As far as I am concerned the data are clearly presented, the paper is well written and the findings are novel and relevant. The cooperative nature of MCT2 is unexpected and shall prompt a revision of current models of intercellular fueling in brain tissue. Authors are invited to consider the following suggestions, which include the realization of new experiments:

We appreciate the reviewer's positive evaluation of our work and its significance. We have addressed all of his/her comments with significant changes to the manuscript, including new text, experiments, and further discussion.

Major

1. MCT2 versus MCT1. Transport rates in MCT2-transduced cells were only twice that of "negative" control cells, which are rich in MCT1 (San Martin et al., PLoS ONE 8(2): e57712). How contaminated is the transport data by MCT1? One possibility is that approx.. 50% of the activity is MCT2 and the rest is MCT1. In that case, the functional data would be hard to interpret. Alternatively, MCT2 overexpression swamped the cellular machinery (expression, trafficking, etc) and the contribution of MCT1 became negligible, a favorable scenario supported by lower than control rates in several MCT2 mutants (R297D, W18A, W20A, R143A). Ensuring that there is no significant MCT1 contamination is important for the correct interpretation of the data. One possible strategy for that would be to use pCMBS, a non-permeant reagent that inhibits MCT1 but not MCT2. If MCT2 is the main MCT in the membrane, pCMBS will not affect pyruvate uptake. A control assay in "negative" HEK cells should show complete inhibition of pyruvate uptake by pCMBS (San Martin et al., PLoS ONE 8(2): e57712).

We thank the reviewer for his/her important insights.

We agree with the reviewer that the negative control signal is high, with approximately 50% of the total signal. However, we would like to clarify that this is the current standard as soon as the MCT-transduced HEK293 cell is used to study pyruvate transport by MCT. As an example, this system had been used in the study of SLC16A11 with a similar high negative control (Rusu et al., Cell, 170:199-212).

We agree with the reviewer that majority of the lactate transport activity in HEK293 cell are from MCT1 in San Martin's study, which showed that pCMBS inhibits approximately 96% lactate influx of HEK293 cell, using a FRET-based Lactate sensor (San Martin et al., PLoS ONE 8(2): e57712). However, **this doesn't come to the conclusion that HEK293 cells are rich in MCT1.**

First, based on the tissue distribution pattern of MCTs (MCT1 is ubiquitously expressed in almost all cells, MCT2 is moderately expressed in kidney, and MCT3 and MCT4 are not expressed in kidney) (Table 1, Halestrap SP, Molecular Aspects of Medicine, 34:337), it is

reasonable to estimate that the endogenous MCTs in HEK293 cell mainly contain MCT1 and MCT2. Therefore, MCT2 does contribute to the negative control signal.

Second, the Michaelis constant (K_m) values of MCT1 and MCT2 for L-Lactate are 3.5 mM and 0.74 mM (Table 2, Halestrap SP, *Molecular Aspects of Mectidine*, 34:337, and Table 1, Broer et al., *Biochem J*, 341:529). In San Martin's study, 5 mM L-lactate was used in the lactate uptake assay (Figure S3, San Martin et al., *PLoS ONE* 8(2): e57712), which is higher than the K_m values of both MCT1 and MCT2 for L-lactate, indicating that both MCT1 and MCT2 of HEK293 cell are fully operational (their transport activities are closed to their maximum rate (V_{max})) under 5 mM L-lactate. However, considering that the V_{max} value of MCT1 is approximately 20-fold higher than that of MCT2 (Figure 2, Broer et al., *Biochem J*, 341:529), inhibition of 96% lactate influx of HEK293 cell by pCMBS in San Martin's study simply means that HEK293 cell contains approximately half by half of endogenous MCT1 and MCT2.

Third, our data reveal that several MCT2 mutants (R297D, W18A, W20A, R143A) show pyruvate transport activities significantly lower than that of negative control, indicating that MCT2 contributes significantly to the negative control signal. Since these MCT2 mutants would form hetero-dimer with endogenous wild type MCT2, affect not only its own activity, but also the activity of the endogenous wt MCT2 in a dimer, resulting in the dominant negative effect.

Fourth, the K_m values of MCT1 and MCT2 for pyruvate are 1.0 mM and 0.08 mM (Table 2, Halestrap SP, *Molecular Aspects of Mectidine*, 34:337). In our study, 0.4 mM pyruvate was used in the pyruvate uptake assay, which is lower than the K_m value of MCT1, and is significantly higher than that of MCT2. In addition, in the analysis of cooperativity of MCT2, majority of the data were recorded with intracellular pyruvate concentration lower than 1 mM, indicating that under our experimental setting, MCT2 is fully operational, while MCT1 is not. This indicates that endogenous MCT2 contributes significantly to the negative control signal, and explains the negligible contribution of MCT1 in our study.

Taken together, we can conclude that HEK293 cell contains approximately half by half of endogenous MCT1 and MCT2. And the contribution of MCT1 becomes negligible under our experimental setting.

We appreciate the reviewer's suggestion to use pCMBS, a non-permeant reagent that inhibits MCT1 but not MCT2. However, we encountered a technical hurdle to order pCMBS. SIGMA and MCE, the two vendors with pCMBS in their catalog, both replied that pCMBS was out of stock. And since few people order pCMBS in the last few years, it would take at least half year for pCMBS back in stock if we place order.

2. MCT2 cooperativity. Estimation of efflux parameters from the time course of pyruvate depletion rests on some assumptions that need verification. It must be shown that extracellular pyruvate is fully removed from the vicinity of cells, otherwise the second phase of slow depletion could be the trivial result of incomplete pyruvate washout. Please provide perfusion rates and volume of the chamber, and demonstrate fast and complete removal, for example by imaging an extracellular dye under the same perfusion regime. Also, it is not clear how reproducible are some measurements. How many times was the experiment in Figs. 1e and 1f carried out?

We used a rapid solution changer with a gravity-driven perfusion system (RSC-200, Bio-Logic) to apply buffer exchange. The perfusion rate was about 1.2 ml/min for all experiments and the volume of the chamber is 50 ml. In the experiments, the cell to be imaged was placed right in front of the outlet of perfusion tube to ensure rapid and complete change of extracellular solution. The diameter of the perfusion outlet is 860 μm , which is much larger than the size of a HEK293 cell (~ 20 μm in diameter). Therefore, the cell was well covered by the perfusion efflux. The experiment in Figs 1e and 1f had been repeated 3 times. We would like to clarify that the same system has been routinely used in William N. Zagotta's lab in University of Washington to perfuse cyclic nucleotides to activate CNG channels, and in Jie Zheng's lab in University of California Davis to perfuse capsaicin to activate TRPV1 ion channel. Our co-corresponding author, Dr. Fan Yang, had been trained in Jie Zheng's lab for ten years. By using this perfusion system, he successfully studied the ligand gating processes where capsaicin and peptide toxin RhTx2 activated TRPV1 channel, respectively (Yang et al., *Nature Chemical Biology* 2015; Yang et al. *Nature Communications* 2015 and 2018). Therefore, we believe that our perfusion system is suitable for the current study.

3. MCT2 cooperativity. Could the slow phase of sensor decay be due to faster bleaching of one of the chromophores? Please provide data demonstrating that bleaching is not significant under your illumination conditions.

We would like to clarify that the chromophore slowly decays due to bleaching. We calibrated the decay by calculating the baseline drift with the first or last 50-80 seconds of each trace.

The bleaching is not significant. Or else, we won't be able to distinguish the curve of AC-R155858 treated cells and that of the AR-C155858 untreated cells (Fig. 1a). We would also like to clarify that the same system had been used in the study of SLC16A11 (Rusu et al., *Cell*, 170:199-212).

4. MCT2 saturation. Reportedly, the zero-trans K_m of pyruvate uptake via MCT2 is in the order of 50 μM (various works by Halestrap and colleagues). However, the current study does not show saturation of zero-trans pyruvate efflux in the range 0-1.6 mM (Suppl. Fig. 2C), which points to a K_m higher than reported for native MCT2. Why is this? Is MCT2 asymmetric? Or is this explained by the fact that the transporter was expressed without its chaperone embigin? Do HEK cells express embigin? Please discuss these issues in relation to relevant MCT2 studies in oocytes by Broer and Deitmer (*Biochem J.* 1999 Aug 1;341 (Pt 3):529-35.; *J Biol Chem.* 2011 Aug 5;286(31):27781-91)

We thank the reviewer for the outstanding question. The K_m values of MCT2 for pyruvate is 0.08 mM (Table 2, Halestrap SP, *Molecular Aspects of Medicine*, 34:337). Therefore, in the range of 0.08-1.6 mM, the transport activity of MCT2 becomes saturated, towards its maximum rate (V_{max}). However, we would like to clarify that MCT2 operates equally well in either direction with the transport rate and direction determined by a combined chemical gradient of proton and transportable monocarboxylates. The relationship between the influx and efflux kinetics is defined by the Haldane equation $\{(V_{\text{max}}/K_m)_{\text{influx}}=(V_{\text{max}}/K_m)_{\text{efflux}}\}$ (Halestrap and Price, *Biochem J.* 343:281-299, 1999). This explains why we didn't observed saturation of pyruvate efflux in the range 0-1.6 mM (Suppl. Fig. 2C), as it is also determined by the chemical gradient. Indeed, when we define MCT2 transport activity as the pyruvate efflux rate versus the intracellular pyruvate concentration, the pyruvate dose-response curve (Fig. 1f, S2d) clearly

show saturation of MCT2 transport activity, when the intracellular pyruvate concentration is higher than 0.08 mM.

To address whether embigin is required for the intracellular trafficking of MCT2 to the plasma membrane, we performed cell surface expression analysis with confocal microscopy by tagging a mEGFP to MCT2. The confocal images clearly show the localization of mEGFP tagged MCT2 to the plasma membrane in HEK293 cells (Supplementary Fig. 1a). Similar to our observation, Klier et al. showed that MCT2 alone can localize to plasma membrane (Fig. 1D, J Biol Chem. 2011 Aug 5;286(31):27781-91). In addition, our data also show that MCT2 alone has the function to transport pyruvate (Fig 1a). Moreover, our unpublished data revealed that embigin is unlikely important for MCT2, while basigin is indeed important for MCT1. The yield of MCT1 dramatically increases when co-expressed with basigin. And MCT1 and basigin tightly interact, being purified as a complex. However, the yield of MCT2 remains similar when co-expressed with embigin. And we were unable to purify embigin-MCT2 complex.

We appreciate the reviewer's suggestion of the two relevant MCT2 studies, we have discussed these issues and have cited the two references.

5. Fluorescent sensor calibration. Please provide a detailed protocol whereby fluorescence data was converted into concentration and illustrate it with an example. This may be included in a supplementary figure.

HEK293T cells co-transfected with pyronic and MCT2 were first bathed in 10 mM pyruvate until equilibration, and were then exposed to buffer without pyruvate. Given that it takes time for the extracellular pyruvate concentration to reach 0 mM after perfusion, the data before 20 seconds since perfusion were not included. For data analysis, baseline drift due to fluorescent bleaching was calculated with the data of the first 50 seconds, or the last 80 seconds (for pyruvate efflux) of each trace. Baseline drift correction with linear regression of initial trace or final trace was performed in the Igor Pro software version 5.05 (WaveMetrics) or OriginPro 2019 (OriginLab). R_0 is the fluorescence ratio of pyronic in the absence of pyruvate and ΔR_{\max} is the difference between R_0 and the maximum ratio estimated in 10 mM pyruvate. The value of the ratio at each time point minus R_0 , ΔR , was transformed into pyruvate concentration (mM) using the measured ΔR_{\max} and the K_D estimated *in vitro* (107 μ M, Fig 2B, Plos One, 2014, 9:e85780). Detail information can be seen in supplementary Fig. 2.

6. Page 12, line 6. According to a recent re-evaluation of MCT4 kinetics using the same pyruvate biosensor, its K_m for pyruvate is 4 mM, not > 150 mM, as previously thought. Thus, the affinity of MCT4 for pyruvate is "only" 100 times lower than that of MCT2, not 2000 times. (<https://www.biorxiv.org/content/10.1101/586966v1>). In the same study the K_m of MCT4 for lactate was measured at 1 mM, which is in the same range of the K_m of MCT1 for lactate. Thus, Ser355 may be key for pyruvate selectivity but not for lactate selectivity. The same article illustrated the perils of estimating MCT parameters using pH.

We appreciate the reviewer's reminding. We have corrected the text and cited the reference.

7. Potential interference by pyruvate metabolism. In addition to efflux, the depletion of pyruvate upon extracellular removal may in principle be affected by other pyruvate sinks, including mitochondria, LDH and aminotransferases. However, such interference would have caused slower accumulation relative to

depletion, which was not observed. Thus, the symmetry between speeds of uptake and depletion suggests that transport dominates. This should be discussed.

We agree with the reviewer that MCT2-transduced HEK293T cell is an imperfect system to study pyruvate transport by MCTs. And there exists potential interference from pyruvate metabolism and other transportable monocarboxylates.

First, in addition to efflux, the cytosolic pyruvate of HEK293T cells has two additional outcomes, being transported into mitochondria as substrate for tricarboxylic acid (TCA) cycle by mitochondrial pyruvate carriers (MPCs), or being chemically modified by enzymes, such as lactate dehydrogenase (LDH) and aminotransferases.

Second, as a proton-linked monocarboxylate transporter, MCT2 is a co-transporter (symporter), with each transport cycle involving a monocarboxylate and a proton. MCT2 transports substrate and proton in both directions, with the transport rate and direction determined by a combined chemical gradient of proton and all transportable monocarboxylates, including pyruvate, L-lactate, D- β -hydroxybutyrate, acetoacetate, α -ketoisocaproate, α -ketoisovalerate, etc.

Third, among the many transportable monocarboxylates of MCT2, the FRET-based pyruvate protein sensor only monitors the pyruvate concentration.

However, in our designed experiment, all transportable monocarboxylates of MCTs are transported outward, their efflux rates and their intracellular concentrations, are likely proportional to each other. In this perspective, the pyruvate release data reflect an overall monocarboxylate transport by MCTs. In this situation, the interference becomes negligible, allowing us to calculate a Hill coefficient (n value) about 1.6. Moreover, the activity-concentration curve of R143A MCT2 reveals a Hill coefficient of 1.0, indicating loss of cooperativity, further demonstrating the negligible interference.

We have discussed this in text.

Minor

8. Page 5 line 5, it should read “regenerating cytosolic NAD⁺ from NADH...”

This sentence has been corrected as suggested.

9. Page 6 line 4, the difference seems larger, e.g. about 100%?

This sentence has been corrected as suggested.

10. Page 6 line 10-12. Highly glycolytic cells accumulate lactate, not pyruvate. Please rephrase.

To avoid confusion, we remove “to mimic the situation of high monocarboxylate concentration in glycolytically active cells” as suggested.

Reviewer #2:

In the manuscript: “Cooperative transport mechanism of the human monocarboxylate transporter 2” the authors describe the cryo-EM structure of the human monocarboxylate transporter (MCT2) in inward open conformation and analyzed its pyruvate transport activity. MCT2 is a solute carrier (member of the SLC16 family) with a typical homodimeric assembly. The essential outcome of the transport assay is the finding that MCT2 can rapidly react to changing monocarboxylate reactions necessitating strong cooperative effects between the individual monomers. The cryo-EM structure is resolved at 3.8Å resolution with solid quality, which allowed for model building. Finally, a computational approach was used to identify residues that are potentially involved in substrate binding. Point mutations of those residues led to a decreased transport activity.

We appreciate the reviewer’s positive evaluation of our work and its significance.

I do not feel confident to ultimately judge the quality of the biochemical and transport assays; but the EM-density looks as expected at this resolution. Unfortunately, 3.8Å is nowadays on the lowish side for cryo-EM of alpha-helical membrane proteins and I wonder whether this could not be improved a bit further to solidify the model and provide a final density. Consequently, higher resolution would also alleviate any potential concerns about substrate binding and the involved residues.

As the EM-micrographs look good and the processing seems to have worked comparably straight forward I am wondering why the authors did not further push the resolution to make the manuscript much stronger. Also, the number of particles obtained (~100k) for the good class should suffice to provide an improved resolution already.

We appreciate the reviewer’s suggestion. However, we would like to clarify that using cryo-EM to determine the structure of MCT2 is challenging due to two reasons. First, MCT2 dimer is small, with a low molecular weight of 104 kDa. Second, MCT2 doesn’t have extracellular and intracellular domain. The purified MCT2 is surrounded by detergent, which made 2-D and 3-D classifications difficult. Further 3D classifications could not improve resolution or map quality of the final 3D reconstruction. We had been trying hard, 3.8 Å is already so far what we can get.

engSpecific comments:

Page 8 line 1: dimer bury an extensive interface of 5,147Å²

Could the authors please doublecheck, whether this is really true? From SF7 the interface appears to be confined to a much smaller area. In any case the number is way to “accurate” at 3.8 Å resolution. I would suggest to round it up.

We appreciate the reviewer’s suggestion. We have double-checked and the interface was correctly calculated with PISA in CCP4. Given the low resolution, we have round it up to “5,100 Å²”.

The manuscript requires some polishing in the wording to improve readability, before publication can be considered.

We thank the reviewer’s suggestion. We have significantly modified the manuscript based on all three reviewers’ comments.

Reviewer #3:

Zhang et al. show a cryo-EM structure of a human monocarboxylate transporter of the SLC16 family, MCT2. After the recent publication of the crystal structure of a bacterial MCT homolog, this first human

structure is highly appreciated - even though a somewhat limited resolution of 3.8 Å was achieved and parts of the structure are missing to due flexibility. Still, the structure exhibits the inside-open conformation, i.e. the complement to the outside-open bacterial structure. The authors combine their structural data with biochemical transport assays of wildtype and mutant MCT2 to foster their structure-derived hypotheses on cooperativity of the two protomers in the dimeric MCT2 complex, and on a protonation site that they make responsible for secondary active, proton-coupled transport of pyruvate. However, unfortunately, the functional data are not convincing.

We appreciate the reviewer's positive evaluation of our work and its significance. We have addressed all of his/her comments with significant changes to the manuscript, including new text, experiments, and further discussion.

Specifically:

Major points:

1. The MCT2 constructs were expressed in a mammalian cell system with a large background pyruvate transport activity. Accordingly, expression of MCT2 increased the transport rates by a small factor of around 1.5 over background. This level might suffice to conclude on general transport functionality but appears too low to attribute true differences to certain mutants. The study fully lacks quantification of the transporter of interest at the plasma membrane. Since the number of transporters directly determines the transport rate this is essential. One cannot know whether the presented rates are directly linked to structural properties of the MCT2 variants or to expression levels.

We thank the reviewer for his/her important insights. We performed western blot to quantitatively detect the expression level and didn't observed significant differences between wild type MCT2 and its mutants (Supplementary Fig. 10b). In addition, we also confirmed the localization of mEGFP tagged MCT2 and its mutants to the plasma membrane by confocal imaging. Compared with wild type MCT2, MCT2 mutants, especially those showing dominant-negative effect (W18A, W20A, R143A), localize to plasma membrane without significant differences on cell surface localization (Supplementary Fig. 10c). These data confirm that the reduced transport rates are directly linked to structural properties of the MCT2 variants, instead of their expression levels.

2. The ms does not mention the accessory single transmembrane proteins basigin or embigin that are required for the intracellular trafficking of the MCTs to the plasma membrane. Altered trafficking due to affected interaction of the accessory protein with the expressed MCT2 variant would also act on the measured transport rates, i.e. a second reason for determining the plasma membrane portion of MCT2.

To address whether embigin is required for the intracellular trafficking of MCT2 to the plasma membrane, we performed cell surface expression analysis with confocal microscopy by tagging a mEGFP to MCT2. The confocal images clearly show the localization of mEGFP tagged MCT2 to the plasma membrane in HEK293 cells (Supplementary Fig. 1a). Similar to our observation, Klier et al. showed that MCT2 alone can localize to plasma membrane (Fig. 1D, J Biol Chem. 2011 Aug 5;286(31):27781-91). In addition, our data also show that MCT2 alone has the function to transport pyruvate (Fig 1a). Moreover, our unpublished data revealed that embigin is unlikely important for MCT2, while basigin is indeed important for MCT1. The yield of MCT1 dramatically increases when co-expressed with basigin. And MCT1 and basigin tightly interact, being purified as a complex. However, the yield of MCT2 remains similar when co-expressed with embigin. And we were unable to purify embigin-MCT2 complex.

3. The assay readout is rather indirect. By using a FRET-based pyruvate protein sensor, the authors obtained data on the pyruvate concentration in the cytosol at the respective observation time points. This concentration, however, is not only altered by transport via the heterologously expressed MCT2 protein but, at the same time, by any cellular process that affects pyruvate concentration by metabolism (lactate dehydrogenase) and alternative transport pathways (endogenous MCT isoforms, and, probably highly relevant, by mitochondrial pyruvate transporters). Even the "pyronic" sensor itself may falsify the observed transport rates at pyruvate concentrations below 100 μM due to its own affinity to pyruvate (K_D of 107 μM !). Therefore, the conclusion on cooperativity of the MCT2 dimer is not justified; effectively, if cooperativity is given, it is a result of many involved components that cannot be resolved with this assay.

We agree with the reviewer that MCT2-transduced HEK293 cell is an imperfect system to study pyruvate transport by MCTs. And there exists potential interference from pyruvate metabolism and other transportable monocarboxylates.

First, as the reviewer pointed out, the cytosolic pyruvate of HEK293T cells has three major outcomes, being exchanged with extracellular side by MCTs, being transported into mitochondria as substrate for tricarboxylic acid (TCA) cycle by mitochondrial pyruvate carriers (MPCs), or being chemically modified by enzymes, such as lactate dehydrogenase (LDH) and aminotransferases.

Second, as a proton-linked monocarboxylate transporter, MCT2 is a co-transporter (symporter), with each transport cycle involving a monocarboxylate and a proton. MCT2 transports substrate and proton in both directions, with the transport rate and direction determined by a combined chemical gradient of proton and all transportable monocarboxylates, including pyruvate, L-lactate, D- β -hydroxybutyrate, acetoacetate, α -ketoisocaproate, α -ketoisovalerate, etc.

Third, among the many transportable monocarboxylates of MCT2, the FRET-based pyruvate protein sensor only monitors the pyruvate concentration.

However, in our designed experiment, all transportable monocarboxylates of MCTs are transported outward, their efflux rates and their intracellular concentrations, are likely proportional to each other. In this perspective, the pyruvate release data reflect an overall monocarboxylate transport by MCTs. In this situation, the interference becomes negligible, allowing us to calculate a Hill coefficient (n value) about 1.6. Moreover, the activity-concentration curve of R143A MCT2 reveals a Hill coefficient of 1.0, indicating loss of cooperativity, further demonstrating the negligible interference.

Regarding the reviewer's concern about pyronic, we would like to clarify that the dissociation constant (K_D) value of 107 μM allows pyronic to respond to pyruvate between 10 μM and 1 mM with a linear relationship (Fig 2B, Plos One, 2014, 9:e85780).

4. One possibility to resolve this issue could be to determine cooperativity in the inward direction. In their assay, the authors maximally loaded the cells with pyruvate and then monitored the changes in the pyruvate release rates with decreasing concentrations over time (with the complications of alternative routes laid out above). It should be possible to determine the uptake rates, in turn, at varying external

pyruvate concentrations; of course this direction of transport would be opposite to the primary physiological pyruvate transport via MCT2. What about lactate transport, i.e. the actual MCT2 substrate?

We thank the reviewer for his/her suggestion. However, we would like to emphasize that HEK293T cell is an imperfect system to study pyruvate transport by MCTs, and to show that monitoring the changes in the pyruvate uptake rates is much more complicated than that in the pyruvate release rates.

As we have mentioned above that MCT2-transduced HEK293 cell is an imperfect system to study pyruvate transport by MCTs. And there exists potential interference from pyruvate metabolism and other transportable monocarboxylates.

In our designed experiments, we first bathed the cells in 10 mM pyruvate, which is significantly higher than the regular cytosolic pyruvate concentration (approximately 0.1 mM), and would result in the influx of pyruvate and the efflux of other transportable monocarboxylates of MCTs until equilibration. The higher intracellular concentration of pyruvate would result in higher concentrations of other monocarboxylates, such as L-lactate, which is converted from pyruvate by lactate dehydrogenase. In the equilibrium state, the combined chemical gradient of proton and all transportable monocarboxylates is close to zero. We then exposed the cells to 0 mM pyruvate, which would result in the efflux of all transportable monocarboxylates, including pyruvate. Given that the FRET-based pyruvate protein sensor only monitors the pyruvate release, the data only partially reflect the monocarboxylate transport by MCTs. However, since all transportable monocarboxylates of MCTs are transported outward, their efflux rates and their intracellular concentrations, are likely proportional to each other. In this perspective, the pyruvate release data reflect an overall monocarboxylate transport by MCTs.

Next we consider the suggested experiment in the inward direction. If we first bath the cells in low concentration of pyruvate, such as 0 mM, which would result in the efflux of all transportable monocarboxylates of MCTs until equilibration. We then expose the cells to higher concentration of pyruvate, such as 10 mM, which would result in pyruvate influx, and however, the efflux of other transportable monocarboxylates of MCTs. Pyruvate influx raises the intracellular concentration of pyruvate, resulting in higher concentrations of other monocarboxylates, lower pyruvate influx rate, higher efflux rates of other transportable monocarboxylates of MCTs and a very complex transport kinetics. In this perspective, the pyruvate uptake data will be very complex and unlikely reflect monocarboxylate transport by MCTs.

5. From their structure analysis and modelling, the authors identified aspartate-293 as a potential protonation site close to the (equally modelled) substrate binding site of MCT2. This residue is made responsible for proton binding and the co-transport property. If this was true and the proton transport mechanism would be based on Asp293, mutation to a neutral asparagine should completely abolish pyruvate transport. Yet, pyruvate transport remained unaltered in the D293N mutant (Fig. 4e) excluding Asp293 from having that impact. Still, the authors conclude from slight, hardly significant changes in the degree of a concurrently observed pH shift (Fig. 4f) that Asp293 is a key residue, which is not conclusive.

MCT2 belongs to the major facilitator superfamily (MFS). Similar to MCT2, many MFS transporters are substrate proton symporters. Although the translocation of proton and substrate are generally believed to be coupled, several MFS transporters, such as XylE and GlcP, have been observed that neutral mutation on protonation site (D27N in XylE, or D22N in GlcP) led to elimination of proton-dependent active symport, but not counterflow activity. We had similar observation on MCT2, leading us to identify Asp293 as a potential protonation site. How the translocation of proton and substrate are decoupled in MCT2, as well as that observed in XylE and GlcP, remains unknown.

We agree with the reviewer that it is not conclusive enough, and therefore tune down our conclusion by using “potential proton-binding residue”.

Minor points:

6. How do the authors explain the differences in the pyruvate uptake levels in the equilibrium state in Fig. 1 (not the different rates)?

We are using HEK293T cell for pyruvate uptake assay. The cytosolic pyruvate has three major outcomes, being exchanged with extracellular side by MCTs, being transported into mitochondria as substrate for tricarboxylic acid (TCA) cycle by mitochondrial pyruvate carrier (MPC), or being chemically modified by enzymes, such as lactate dehydrogenase (LDH) and aminotransferases. In our pyruvate uptake assay in Fig. 1a, we raised the extracellular pyruvate concentration to 0.4 mM, which is significantly higher than the regular cytosolic pyruvate concentration (approximately 0.1 mM), and will result in pyruvate influx. In the equilibrium state, the cytosolic pyruvate concentration is determined by a balance between pyruvate influx from extracellular side by MCTs, pyruvate influx into mitochondria by MPCs, and pyruvate modification by enzymes. It is reasonable to assume that the control cells and the MCT2-overexpressed cells contain similar amounts of MPCs and pyruvate modification enzymes. The MCT2-overexpressed cells have a higher influx rate than that of control cells, which then shifts the cytosolic pyruvate to a higher concentration in the MCT2-overexpressed cells than that in the control cells in the equilibrium state.

7. Cryo-EM is a method that allows for depicting different conformational states via class summation and selection of different classes. Why was only the inside-open conformation picked up in the analysis?

This depends on the free energy states of different conformational states. If the energy differences between different conformational states are low, it becomes easier to depict different conformational states via class summation and selection of different classes, and vice versa. For MCT2, the dominant state we observed is the inward facing conformation, suggesting that it is the lowest energy state for MCT2.

In conclusion, again, the novel structure in its conformation is particularly helpful. Yet, the deduced structure-function relationships regarding cooperativity and protonation appear not conclusive in the current ms.

REVIEWERS' COMMENTS:

Reviewer #1 (Remarks to the Author):

I think authors have addressed the issues regarding functional characterization of the transporters to a reasonable extent, including new experiments. A pity that pCMBS assays were not possible. Structural aspects are beyond my expertise.

Reviewer #3 (Remarks to the Author):

The authors have adequately addressed most of the comments in particular those concerning protein expression levels and localization. At the time, the authors have explained in their rebuttal letter that the HEK293 expression system is not fully suitable for all conclusions drawn. In the light of this, my recommendation is to appropriately use "might"/"may"/"could" in the respective parts of the text to properly indicate uncertainty. Otherwise, the reviewer is satisfied with the revision.

Response to reviewers' specific comments:

Reviewer #1 (Remarks to the Author):

I think authors have addressed the issues regarding functional characterization of the transporters to a reasonable extent, including new experiments. A pity that pCMBS assays were not possible. Structural aspects are beyond my expertise.

We thank the reviewer for his/her appreciation.

Reviewer #3 (Remarks to the Author):

The authors have adequately addressed most of the comments in particular those concerning protein expression levels and localization. At the time, the authors have explained in their rebuttal letter that the HEK293 expression system is not fully suitable for all conclusions drawn. In the light of this, my recommendation is to appropriately use "might"/"may"/"could" in the respective parts of the text to properly indicate uncertainty. Otherwise, the reviewer is satisfied with the revision.

We thank the reviewer for his/her appreciation. We have modified the main text accordingly.